# Influence of Rural Development of River Tourism Resources on Physical and Mental Health and Consumption Willingness in the Context of COVID-19

Hsiao-Hsien Lin [1], Kuo-Chiang Ting [2], Jen-Min Huang [3], I-Shen Chen [4] and Chin-Hsien Hsu [4,*]

1   School of Physical Education, Jiaying University, Meizhou 514015, China; chrishome12001@yahoo.com.tw
2   Department and Graduate Institute of Physical Education, University of Taipei, Taipei 10048, Taiwan; kelven0923@icloud.com
3   Department of Physical Education, National Pingtung University, Pingtung City 90003, Taiwan; jenmin650317@gmail.com
4   Department of Leisure Industry Management, National Chin-Yi University of Technology, Taichung 41170, Taiwan; ishenc@ncut.edu.tw
*   Correspondence: hsu6292000@yahoo.com.tw

**Abstract:** This study took the Three Gorges Dam as an example and discussed the influence of river regulation decisions on the sustainable development of surrounding villages. The study used mixed research methods, snowball sampling, and convenience sampling to obtain samples. The questionnaire samples were analyzed by basic statistical tests, *t*-test, and structural equation modeling (SEM). The respondents' opinions were collected through semi-structured interviews and finally the results were discussed by multivariate analysis. The findings were that even though the villages were not well developed in terms of economy, environment, and natural ecology, as long as the community security could be stable, the living could be safe and convenient, people's daily life patterns and leisure behaviors could be maintained, and people could stabilize their minds and emotions and maintain physical and mental health in order to meet their living needs and reduce the burden. There would be time and funds to invest in leisure, tourism activities, and consumption behavior. If the above consumption patterns are continued, people will gain positive perceptions, stimulating people's willingness to invest in property purchases or to make travel plans again.

**Keywords:** river improvement; environmental management; different stakeholders; consuming behavior; rural sustainability

## 1. Introduction

Rivers have abundant freshwater resources, their general drainage areas are extensive, and alluvial plains are fertile, nourishing and enriching the ecology [1,2]. Human beings have rich experience and a long history [3–5] of using the advantages and foundation of river water resources to develop fishing, animal husbandry, farming, and other industries to maintain their needs, build settlements, develop cities, and create a safe and sustainable living and living environment. It can be seen that safe and stable water resources will help improve the development of human beings and villages.

The Yangtze River is the third largest river in the world, with a total length of 6300 km. The terrain is high in the west and low in the east, forming three ladders. The trunk spans 11 provinces in China, and the tributaries cover 19 provinces [6]. However, due to the subtropical monsoon climate in the Yangtze River Basin, the annual rainfall is abundant during the rainy season. The upstream development is not properly developed, resulting in serious water and soil loss, siltation in waterways, and increasing riverbeds [7,8]. Under climate change, the frequency of extreme rainfall is greatly increased so that the rainfall is severe, the water channel is curved, the terrain around the river channel is undulating, and the water level between the high tide and the river channel can be up to 10 m or more; all

these impact on the human and ecological environment [9–11]. After 16 years of planning, the local government has built and improved the Three Gorges Dam water conservancy project [12]. The dam has a water source area with a total length of 193 km [13], a maximum water storage capacity of 39.3 billion cubic meters, and a power generation of 100 billion kegs [14]. The dam is a multi-functional reservoir water source area with the benefits of flood control, flood resistance, power generation, transportation, irrigation, shipping, tourism, and water source scheduling [13,14], providing diverse and rich leisure and tourism resources, and providing development and utilization of surrounding villages [15]. The surrounding villages have invested various funds and constructions and used the existing cultural, natural environment, and ecological resources to develop leisure and tourism activities such as historical sites, culture, river cruises, ferries, fishing, and river fish festivals [15].

The main purpose of constructing the Three Gorges Dam is to solve the flooding [12] and create safe village living and living conditions. However, people are more looking forward to developing leisure or tourism activities, promoting local development, and improving the quality of life and health [15–18]. Although development decisions can improve local predicaments [19], they can also assist in developing leisure and tourism activities and promote local development [20]. Scholars believe that public construction or tourism development decisions interfere with the village economy, society, environment, and natural environment [18–22]. In terms of economy, it can improve industrial construction, livelihood price, and village development [21]. However, in terms of society, village construction, industrial human resources, and public security management will be challenged [19,22]. In terms of environment, the current village environment and public health management may be disturbed by development [18]. In terms of the natural environment, the existing natural ecology and environmental appearance will be destroyed due to decision-making and development [23,24]. With the impact of the epidemic, the current situation of the village environment and sanitation maintenance is easily disturbed [19]. As a result, the effectiveness of decision-making and management will decline, environmental risks will increase, people's physical and mental health will be threatened, and the willingness to live or travel will be affected [18,19].

With the rise of tourism-related industries and the improvement of public construction, the current situation of tourism development in local villages has gradually improved and has been favored by the public [13–15]. According to statistics, the population of the surrounding villages has grown to 35.8137 million people, and it can attract about 72.23 million tourists every year, creating business opportunities of USD 9.2669 billion [17]. It can be seen that the benefits of the construction of the Three Gorges Dam on the Yangtze River will be of great help in improving the development of surrounding villages. However, decision-making often changes due to various factors, the expected goals and actual promotion results often have positive and negative effects [21,25], and its changes take time to verify [19]. Scholars believe that the true appearance of decision-making development results can be known through tourists' travel experiences [20,26,27]. Using the personal experience and feelings of residents can provide the real results of changes in the development status of the village and the surrounding natural environment [19,28]. Although both strategies use existing resources to meet individual life or travel needs, can improve individual physical and mental health status, can induce people's willingness to travel again [29–36], and can even generate the idea of buying lands or houses, residents and tourists still have different needs and experiences [21]. Therefore, some scholars suggest that if we want to understand the influence of the current development of villages and natural ecology, we should focus on whether the expectations of residents and tourists can be met. Therefore, the perspectives of different stakeholders are analyzed in this manuscript, and we can obtain answers from the perspectives of residents and tourists [19–22,26–28,37–40].

Furthermore, whether the government creates a safe living atmosphere and a healthy leisure environment and achieves people's expectations is not an issue that can be interpreted purely from the perspective of decision makers [21,29], nor can it simply listen to

the opinions of tourists [30], rely on the number of tourists, or be measured by the total amount of consumption [31]. Although the decision is to improve the local plight, creating a safe living atmosphere and a healthy leisure environment is the goal [21,32]. The ultimate goal of local promotion of leisure, tourism facilities, and industrial activities is to obtain the recognition of tourists so that they can continue to travel and consume, thus bringing business opportunities [33,34]. When the development is on track, it can improve the living standards of residents, build a safe leisure environment and living atmosphere, enhance the willingness to live, increase the manpower for industrial development [33,35], and finally achieve the goal of sustainable urban development [36]. Therefore, if we obtain residents' opinions from multiple considerations, refer to tourists' suggestions, and then analyze from the different perspectives of the two parties [19,22,33,40], we may obtain more realistic answers.

Finally, we have read the relevant literature on water resources, rivers, and reservoirs in recent years and found many current research topics on rivers, water resources, fresh water, dams, and water source areas. However, most of the topics discussed are tourism development and impacts [19,22–25], freshwater intake sources and strategies [41], water quality and monitoring [42], carbon emissions [43], hydrology [44], water pollution [45], and marine and freshwater ecosystems [46]. In addition, the current research topics focusing on the Yangtze River only focus on air pollution [47], water pollution [48], energy utilization rate [49], urban land development and utilization rate [50], cruise tourism development [51], and residents' environmental literacy and decision-making participation [52]. Most of them are simply discussed from a single perspective, such as that of residents and tourists [51–53], but there is no hybrid research method, and most are based on the perspective of consumer willingness to buy property, revisit, or the two perspectives combined to discuss together. There are still gaps in the research topics related to consumer behaviors such as dams, villages, natural ecology and environment, and human habitation and tourism. Therefore, we believe that we can take the Three Gorges Dam of the Yangtze River as a case, use various research methods to obtain multi-faceted information, summarize the research information through the processes of classification, induction, and ranking, and cross-compare the information [54–56]. Then, the significance of the research information can be discussed by the multivariate inspection analysis method; the influence of the river dam facilities on the surrounding villages, the natural environment, and the current ecological development can be analyzed; and more in-depth answers can be obtained [57–59].

The main purpose of this research is to reflect people's confidence in the effectiveness of decision-making from the willingness to live. The willingness to revisit reflects the tourists' recognition of the decision-making development effect. Then, we can understand the impact of the construction of water dams on the village, the natural environment, ecological development, and the influence on people's physical and mental health cognition, consumption, willingness to live and travel, and behavior. Based on the analysis results, it is expected that we can put forward decision-making suggestions for the sustainable development of river water conservancy projects and urban development to fill the research gaps. This is the value and contribution of this research.

## 2. Literature Discussion

### 2.1. Physical and Mental Health

Physical and mental health refers to an individual's physical, psychological, and social aspects of reaching a state of well-being [60,61]. It presents the actual situation of individual perception through self-assessment tests [62], and it can be an analytical method of self-perceived assessment [63]. The higher the health risk, the greater the influence on individual behavioral decisions [64].

Investigating individuals' physical and mental health through personal feelings can show the impact of the current environment on people [65]. Some scholars believe that it can usually be divided into three levels: spirit, psychology, and attitude [64–67]. It can be used to explain anxiety, ability, enthusiasm, headache, insomnia, abdominal pain,

abnormal diet, stomach pain, and thoughts of death [68,69], with other feelings waiting to be confirmed. Moreover, studies have confirmed that individuals with different identities have different opinions on their physical and mental health. Their physical and mental health quality will affect their consumption intentions and follow-up judgments [23,64].

Therefore, we believe that if we want to understand a person's physical and mental health, we should evaluate it from three levels: psychology, spirit, and attitude. Then, we can learn about the public's feelings about personal physical and mental health in terms of headache, abdominal pain, anxiety, ability, enthusiasm, insomnia, stomach pain, abnormal diet, and thoughts of death.

### 2.2. Willingness to Consume

Willingness refers to the inner intention generated from individual psychology [60], and behavior transforms into actual external action when the intention appears [70]. The actual consumption behavior can be predicted by the willingness to consume [70]. It can also be assessed through the degree of involvement in purchasing a certain commodity or engaging in specific consumption behavior, whether the current consumption behavior or action meets personal life needs and expected goals [71]. A strong willingness to consume will increase the degree of involvement in consumer behavior [72]. When individuals have sufficient trust in the activities or commodities they are about to participate in, it will enhance consumers' willingness or behavior to consume [72,73]. It can be seen that the willingness to consume can reflect people's intentions to participate in activities or purchase goods, thereby anticipating the decision-making process and evaluating the degree of involvement, trust, and demand for actual actions.

Some scholars believe that living and tourism are consumption intentions and behaviors [74,75]. Consumption willingness can be assessed in terms of continuous consumption, recommending relatives and friends to consume, and sharing experiences [74]. Consumption behavior, part of which is recommending relatives and friends to consume together, can be measured to determine consumption [75]. It is confirmed that there are differences in the views of different stakeholders. The quality of the decision-making effect will likely affect the people's willingness to go to the local area for leisure and tourism activities. The level of willingness to consume affects the degree of involvement in actual consumption behavior [19–23,60,70–75].

Therefore, if we want to understand people's feelings about decision-making effectiveness, we should start with people's perception of life and travel willingness to discuss topics in continuous consumption, sharing experiences, and introducing relatives and friends to consume. We can know people's confidence and recognition of the effectiveness of decision-making.

### 2.3. Cognition of Decision-Making Development

Decision-making refers to a plan formed by an individual or an organization to achieve a certain goal or solve a certain problem [21]. Decision development is accomplished through experience, learning, thinking, analysis, and judgment, and is carried out according to information, design, planning, and execution [26]. When people start to promote decision-making, after people experience, analyze, and judge in person, they will judge and evaluate the decision-making, which is decision-making cognition [22,23,26]. The decision-making mainly aims to improve the local predicament and promote local development [23], biological coexistence, and the sustainable development of resources.. Therefore, if we can capture the opinions of the public after their personal experience and analyze the effectiveness of urban and natural ecological development, we will obtain more appropriate information.

#### 2.3.1. Economic Development

Economic development refers to the interaction process between people and resources and is individuals' consumption behavior and phenomena. It can also refer to urban

development in general [76]. Economic development refers to when human beings have obtained basic survival needs after collecting natural resources. The remaining resources are exchanged for commodities to meet the needs of the masses and achieve the goal of sustainable development of resource diversification [77]. However, due to the increase in population, the gradual scarcity of natural resources, and the impact of natural disasters over time, local economic development will still experience a crisis [23]. Although the government expects to improve the local development dilemma by making decisions [21], the development will have positive and negative effects due to uncertain factors [19–23]. Therefore, analyzing the current economic development situation, it is helpful to improve the problem and achieve the goal of sustainable development [18,19].

Some scholars believe that discussions from the perspectives of industrial construction, the price of people's livelihoods, the overall development quality of villages [18], and people's personal experiences can reflect the real answer to the current state of economic development [19]. Some studies have pointed out that the views of different stakeholders are different, and the decision-making results impact the village's current economic development, thus affecting the willingness to consume [22,23].

Therefore, we believe that if we want to know the effectiveness of economic development, we should start with issues such as industrial construction, the price of people's livelihoods, and the overall development quality of villages, and analyze based on people's personal experience and opinions. We may then know the degree of the impact of decision-making on urban economic development.

### 2.3.2. Social Development

Society refers to people's common living habits, customs, and culture [78]. It can be derived by gathering the same ethnic group or common culture, beliefs, and living habits [33,78]. Social development refers to the consciousness, behavior, interaction mode, or phenomenon that maintains unique characteristics of education, culture, living habits, and other characteristics in a city, region, or country after forming human settlements [79]. However, the unique culture will disappear with different levels of education, cross-ethnic and regional cultural exchanges, and population decline [32]. In addition, existing facilities will become old with time [79], and local social and cultural development will be challenged by factors such as natural disasters and man-made destruction [19]; there will still be positive and negative influences on decision-making development due to uncertain factors [19–23]. Therefore, by analyzing the current situation and predicament of social and cultural development, it is helpful to construct a safe and comfortable living and tourism environment [22,23].

Some scholars believe that discussing the aspects of community building, public security, and industrial human development [18] and considering people's personal experiences can reflect the real answer to the current situation of social development [19]. Some studies have pointed out that the views of different rights stakeholders are different, and the decision-making results have an impact on the social and cultural development of the village, thus affecting the willingness to consume [22,23].

Therefore, if we want to know the effect of social and cultural development, we can start with issues such as community building, public security, and industrial human development and analyze them based on people's personal experiences and opinions. We may then know the degree of the impact of decision-making on urban social development.

### 2.3.3. Environmental Development

In a broad sense, the environment refers to the space required by every piece of land, air, water resources, and various ecologies and species on the Earth [80]. The social environment is the space in which human beings live after artificial improvement or establishment [81]; the natural ecological environment is the material, ecology, and space that have not been artificially changed [82]. When humans use the surrounding artificially improved or natural space, ecology, and environment to absorb their resources,

a consciousness, behavior, interaction mode, or phenomenon that basically satisfies human development and maintains ecological balance [83] is the development of society and the natural environment. However, society and the natural environment are easily destroyed due to natural disasters, diseases, or man-made development, resulting in unbalanced development [79–83]. Although the government expects to improve the local development dilemma by making decisions [29,33], the outcome of that decision-making will still lead to positive and negative changes due to uncertain factors [19–23]. Therefore, by analyzing the development status of urban society and the natural ecological environment, it is helpful to formulate a sustainable development environment [23,30].

Some scholars believe that the social environment can be considered from the overall village environment, public health, and other aspects [23,33]. The natural environment can be discussed regarding water resources, soil, ecology, forest land, and human destruction [19]. Through people's personal experiences, it can reflect the real answer to the development status of society and the natural environment [18,19]. Some studies have pointed out that the views of different stakeholders are different, and the decision-making results have an impact on the development of society and the natural environment, thus affecting the willingness to consume [19,22,23].

Therefore, we believe that if we want to know the development effect of society and the natural environment, society should focus on the overall environment of the village and the development of public health. The natural environment should be cut from water resources, soil, ecology, forest land, and human destruction. Moreover, based on the personal experience and opinions of the people, we may know the degree of influence of the development of the urban society and the natural ecological environment.

## 3. Methods

### 3.1. Research Framework

This study aimed to explore the impact of the Three Gorges Dam on urban development and people's willingness to live, relax, and travel. The study referred to the relevant literature on the Three Gorges Dam [6–15] and the relevant research results on water resources, tourism development, consumption willingness, and urban development [61–79]. We collected data from various aspects, obtained the opinions of people with different identities, and conducted logical derivation through exploration, discovery, and induction [84]. Therefore, the answers could be obtained to meet the expectations of residents and tourists. Figure 1 illustrates the main research framework.

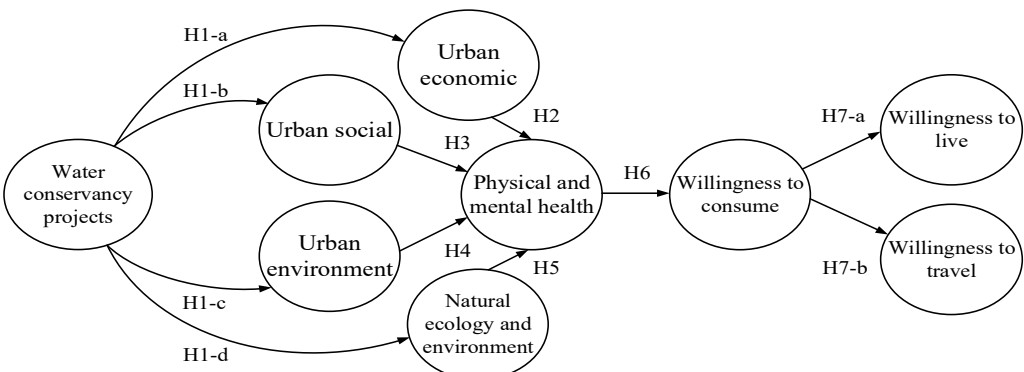

**Figure 1.** Research framework.

### 3.2. Research Hypothesis

Based on the above inferences and research framework planning, we proposed seven research hypotheses.

**Hypothesis 1a.** *The public believes that the influences of water conservancy projects on the development of the urban economy are all consistent.*

**Hypothesis 1b.** *The public believes that the influences of water conservancy projects on the development of the urban society are all consistent.*

**Hypothesis 1c.** *The public believes that the influences of water conservancy projects on the development of the urban environment are all consistent.*

**Hypothesis 1d.** *The public believes that the influences of water conservancy projects on the development of the natural ecology are all consistent.*

Although water conservation projects aim to reduce flood hazards and provide a safe living environment, the promotion of projects can still cause damage to the existing environment [12–18]. Thus, we think the public may feel the same way. Hypotheses 1a to 1d are established.

**Hypothesis 2.** *Village economic development has a positive and significant impact on physical and mental health cognition.*

**Hypothesis 3.** *Village social development has a positive and significant impact on physical and mental health cognition.*

**Hypothesis 4.** *Both natural ecology and environmental development status have a positive and significant impact on physical and mental health cognition.*

**Hypothesis 5.** *The current status of the urban economy, society, environment, natural ecology, and environmental development has a positive and significant impact on physical and mental health cognition.*

Convenient living spaces can improve people's quality of life. A comfortable environment can reduce anxiety [16]. Safe spaces can relieve stress [15]. Adequate leisure can promote health [20]. If the above conditions can be met simultaneously, it can promote people's physical and mental health [15,16,18]. Therefore, we believe that if the Three Gorges Dam improves villages and the natural ecology, people's physical and mental health can also be improved. Hypotheses 2 to 5 are established.

**Hypothesis 6.** *Physical and mental health has a positive and significant impact on willingness to consume.*

Although long-term physical and mental health is the goal that everyone expects [22], it is necessary to meet the conditions of maintaining physical and mental health through consumption. Therefore, when people have positive physical and mental health, they will have consumption awareness, such as life and entertainment. However, when a negative physical and mental health state occurs, there will be consumer awareness, such as medical care [33,35]. Therefore, we believe that self-awareness of physical and mental health has an influence on consumption intention. Hypothesis 6 is established.

**Hypothesis 7.** *Consumption willingness has a positive and significant impact on both living (7-a) and travel willingness (7-b).*

Consumption behavior is a manifestation of satisfying people's physical and psychological needs [70]. When people are able to meet their basic survival needs, they will move to the next stage of consumption [71,72]. Improving living standards, relieving stress, and improving physical and mental health through tourism is the ultimate goal of people's quality of life [85]. Therefore, we believe that when people's physical and mental health improves, it will promote people's consumption awareness or behavior to improve their liv-

ing quality by buying a house or traveling to improve their living standards. Hypothesis 7 is established.

*3.3. Study Subjects, Limitations, and Sampling Methods*

As the researchers, we took the cities surrounding the Three Gorges Dam water conservancy project as the scope. We took local residents and people with local tourism experience as the object. We explored the impact of the Three Gorges Dam project on the development of the village's economy, society, environment, and ecological environment and its impact on physical and mental health, consumption willingness, and willingness to live and re-tour.

The team used the field investigation method to first visit the Three Gorges Reservoir and surrounding villages. The convenience sampling method was used to collect samples of residents' questionnaires; semi-structured interviews were combined with notes, voice recorders, and other tools to collect residents' views on issues. The questionnaire survey method was used to collect samples using an online questionnaire platform. Then, using the snowball sampling method, those who had travel experience to the villages or scenic spots around the Three Gorges Dam on the Yangtze River were invited to accept the survey and the respondents were asked to assist in forwarding the survey to expand the scope and quantity of the sample collection.

However, since the total area of the Three Gorges Dam water conservancy project is about 126 km, and the villages cover a wide area, the researchers took a lot of time on this study. Because the future of the pandemic has not yet become clear, both the surveyors and the respondents were reluctant to take the risk of infection. Although the sampling conditions of the online questionnaire are limited, the uncertainty of respondents' willingness to answer and actual travel experience is relatively high. In addition, the time, manpower, material resources, financial resources, and other constraints of the research team would affect the sample size ratio and research results. Any deficiencies in the number of samples and research results could then become a follow-up research suggestion, and other researchers could try to strengthen it.

*3.4. Research Processes and Tools*

3.4.1. Exploratory Factor Analysis

First, we collected the relevant literature on the Three Gorges Dam water conservancy project [6–15] and determined the topic and direction, and then we discussed the literature related to water resources, physical and mental health, consumption willingness, and urban development [4,17,19] and confirmed the research strategy. After referring to relevant literature and editing the preliminary questionnaire topics, three scholars with backgrounds in decision analysis, environmental management, and public health decision analysis were invited to conduct a content validity check to confirm the content of the formal questionnaire.

The formal questionnaire was divided into seven parts. Background information was distinguished by issues such as identity (resident or tourist) and gender (male or female). All other issues were on a 5-point Likert scale. A score of 5 meant strongly agree, and a score of 1 meant strongly disagree. Issues such as urban development, the natural ecology of rivers, consumption willingness, and behavior were divided into the urban economy (4 questions), urban society (5 questions), urban environment (4 questions), and natural ecological environment (4 questions) after referring to literature such as [18–66]; including physical and mental health cognition (4 questions), willingness to consume (3 questions), willingness to live (3 questions), and willingness to travel (3 questions), a total of 26 questions were set.

One hundred samples were collected in May 2022, and SPSS 26.0 statistical software was used for basic statistical verification. The effective recovery rate of the pre-test questionnaire was 100%. Some scholars have pointed out that when Kaiser–Meyer–Olkin (KMO) > 0.06 and the *p*-value in Bartlett's test is less than 0.01 ($p < 0.01$), it indicates that

the scale is suitable for continuous factor analysis [86]. Moreover, when the coefficient $\alpha$ is greater than 0.60, this indicates that the questionnaire has good reliability [87], and continuous analysis can be retained.

The analysis showed that the KMOs of the urban economy, urban society, urban environment, natural ecology and environment, consumption willingness, living willingness, and tourism willingness were all over 0.06, the significance was 0.001, and the coefficient $\alpha$ were all greater than 0.60. These results meant that each aspect and its issues could continue to be analyzed in succession, as shown in Table 1.

**Table 1.** Questionnaire analysis of the urban economy, urban society, urban environment, natural ecology, environment, willingness to consume, willingness to live, and willingness to travel.

| | KMO | Bartlett ($\chi^2$) | df | $p$ | $R^2$ | Issue | $\alpha$ |
|---|---|---|---|---|---|---|---|
| Urban economic | 0.862 | 1246.02 | 28 | 0.000 * | 84.176% | (Ec-1) Combining characteristics with local industries; (Ec-2) increasing entrepreneurship and employment opportunities; (Ec-3) improving transportation planning; (Ec-4) developing protection policy. | 0.932–0.936 |
| Urban social | 0.864 | 922.583 | 21 | 0.000 * | 81.527% | (S-1) Improve the quality of tourism services; (S-2) increase leisure opportunities; (S-3) public participation in decision-making; (S-4) increase friendly interaction; (S-5) increase security quality. | 0.924–0.929 |
| Urban environment | 0.812 | 733.985 | 15 | 0.000 * | 75.649% | (En-1) More convenient transportation; (En-2) exhaust and noise pollution from automobiles and locomotives increases; (En-3) adequate trash cans; (En-4) adequate toilets; (En-5) landscape and heritage maintenance. | 0.898–0.902 |
| Natural ecology and environment | 0.860 | 791.19 | 10 | 0.000 * | 77.643% | (NEE-1) Turbid river; (NEE-2) soil erosion along the river; (NEE-3) ecological species decline; (NEE-4) ecological habitat change. | 0.922–0.940 |
| Physical and mental health | 0.859 | 1337.224 | 36 | 0.000 * | 80.915% | (PMH-1) Headache; (PMH-2) stomach pain; (PMH-3) abnormal diet; (PMH-4) suicidal ideation. | 0.911–0.923 |
| Willingness to consume | 0.563 | 778.991 | 15 | 0.000 * | 71.53% | (WC-1) Willingness to consume again; (WC-2) recommend relatives and friends; (WC-3) share experience. | 0.851–0.866 |
| Willingness to live | 0.568 | 194.932 | 3 | 0.000 * | 34.306% | (WL-1) Have the idea of buying property and settling down; (WL-2) have the idea of recommending relatives and friends to move in; (WL-3) have the idea of sharing experience. | 0.667–0.713 |
| Willingness to travel | 0.564 | 192.832 | 3 | 0.000 * | 25.859% | (WT-1) Have the idea of revisiting consumption; (WT-2) have the idea of recommending relatives and friends to travel; (WT-3) have the idea of sharing experience. | 0.688–0.728 |

* $p < 0.001$.

### 3.4.2. Confirmatory Factor Analysis

Then, CFA was used to test the reliability and validity of the questionnaire, and the items were revised with a reference (modification indices, M.I.) to make corrections [85] and test the issues of the scale in this study. Those that met the criteria were retained, and others were deleted.

1. Verification of convergent validity

Some studies have shown that the combined reliability (C.R.) and variance extraction (AVE) measures of questionnaire dimensions were used as tests of convergent validity [88]. Therefore, to obtain good convergent validity of the questionnaire, it is recommended that the C.R. value should be greater than 0.6 and the AVE value should be greater than 0.5 [89]. This study conducted convergent validity tests on the dimensions of urban social development, urban environmental development, urban economic development, natural environment and ecological development, consumption willingness, living willingness, travel willingness, and physical and mental health. The factor loadings of all facets were between 0.67 and 0.94; the C.R. value was between 0.73 and 0.94; and the AVE value was between 0.50 and 0.74, which was in line with the convergent validity criteria suggested by scholars [88–90]. This indicated that the convergent validity of the questionnaire in this study was good, as shown in Table 2.

**Table 2.** Confirmatory factor analysis.

| Facets | Indicators | Standardized Load | Unstandardized Load | S.E. | C.R. (*t*-Value) | *p* | SMC | C.R. | AVE |
|---|---|---|---|---|---|---|---|---|---|
| Urban social | S-1 | 0.82 | 1.00 | | | | 0.68 | 0.93 | 0.73 |
| | S-2 | 0.84 | 0.92 | 0.05 | 16.81 | *** | 0.70 | | |
| | S-3 | 0.90 | 1.14 | 0.06 | 18.73 | *** | 0.82 | | |
| | S-4 | 0.85 | 1.04 | 0.06 | 17.15 | *** | 0.72 | | |
| | S-5 | 0.88 | 1.06 | 0.06 | 17.93 | *** | 0.77 | | |
| Urban environment | En-1 | 0.73 | 1.00 | | | | 0.53 | 0.90 | 0.70 |
| | En-2 | 0.89 | 1.19 | 0.08 | 14.50 | *** | 0.80 | | |
| | En-3 | 0.88 | 1.23 | 0.09 | 14.36 | *** | 0.77 | | |
| | En-4 | 0.86 | 1.20 | 0.09 | 14.01 | *** | 0.74 | | |
| Urban economic | Ec-1 | 0.76 | 1.00 | | | | 0.57 | 0.90 | 0.70 |
| | Ec-2 | 0.86 | 1.27 | 0.09 | 14.78 | *** | 0.74 | | |
| | Ec-3 | 0.85 | 1.20 | 0.08 | 14.38 | *** | 0.73 | | |
| | Ec-4 | 0.88 | 1.22 | 0.08 | 15.29 | *** | 0.77 | | |
| Natural ecology and environment | NEE-1 | 0.91 | 1.00 | | | | 0.83 | 0.94 | 0.79 |
| | NEE-2 | 0.91 | 0.99 | 0.04 | 24.49 | *** | 0.84 | | |
| | NEE-3 | 0.92 | 0.97 | 0.04 | 24.72 | *** | 0.85 | | |
| | NEE-4 | 0.83 | 0.82 | 0.04 | 19.48 | *** | 0.70 | | |
| Willingness to consume | WC-1 | 0.77 | 1.00 | | | | 0.60 | 0.85 | 0.65 |
| | WC-2 | 0.86 | 1.08 | 0.07 | 14.58 | *** | 0.74 | | |
| | WC-3 | 0.80 | 0.93 | 0.07 | 12.75 | *** | 0.65 | | |
| Willingness to live | WL-1 | 0.85 | 1.00 | | | | 0.72 | 0.85 | 0.67 |
| | WL-2 | 0.95 | 1.05 | 0.07 | 14.29 | *** | 0.90 | | |
| | WL-3 | 0.63 | 0.58 | 0.05 | 11.35 | *** | 0.40 | | |
| Willingness to travel | WT-1 | 0.70 | 1.00 | | | | 0.49 | 0.84 | 0.64 |
| | WT-2 | 0.93 | 1.26 | 0.11 | 11.85 | *** | 0.87 | | |
| | WT-3 | 0.77 | 1.03 | 0.09 | 11.68 | *** | 0.60 | | |
| Physical and mental health | PMH-1 | 0.77 | 1.00 | | | | 0.60 | 0.91 | 0.73 |
| | PMH-2 | 0.94 | 1.25 | 0.07 | 17.10 | *** | 0.88 | | |
| | PMH-3 | 0.87 | 1.11 | 0.07 | 15.71 | *** | 0.76 | | |
| | PMH-4 | 0.83 | 1.10 | 0.07 | 14.76 | *** | 0.69 | | |

*** $p < 0.001$.

2. Discriminant validity

The discriminant validity of the study was tested using the confidence interval method (bootstrap). First, it was checked whether there was a 1 in the confidence interval of the Pearson correlation coefficient between the questionnaire components in this study to determine whether the questionnaire components were completely correlated. The results showed that the confidence interval between the questionnaire dimensions did not contain 1, indicating that the questionnaire in this study had significant discriminant validity [91], as shown in Table 3.

**Table 3.** Bootstrap correlation coefficient 95% confidence interval table.

| Parameters | | | Estimations | Bias-Corrected | | Percentile Method | |
| --- | --- | --- | --- | --- | --- | --- | --- |
| | | | | Lower Bound | Upper Bound | Lower Bound | Upper Bound |
| Urban social | <–> | Urban environment | 0.81 | 0.68 | 0.89 | 0.68 | 0.88 |
| Urban social | <–> | Urban economic | 0.84 | 0.78 | 0.91 | 0.76 | 0.90 |
| Urban social | <–> | Natural ecology and environment | 0.50 | 0.34 | 0.63 | 0.33 | 0.62 |
| Urban social | <–> | Willingness to consume | 0.65 | 0.47 | 0.78 | 0.44 | 0.77 |
| Urban social | <–> | Willingness to live | 0.00 | −0.13 | 0.11 | −0.12 | 0.12 |
| Urban social | <–> | Willingness to travel | 0.20 | 0.01 | 0.39 | 0.00 | 0.38 |
| Urban social | <–> | Physical and mental health | 0.75 | 0.64 | 0.85 | 0.63 | 0.84 |
| Urban environment | <–> | Urban economic | 0.65 | 0.51 | 0.77 | 0.80 | 0.76 |
| Urban environment | <–> | Natural ecology and environment | 0.69 | 0.51 | 0.81 | 0.47 | 0.80 |
| Urban environment | <–> | Willingness to consume | 0.00 | −0.12 | 0.14 | −0.13 | 0.14 |
| Urban environment | <–> | Willingness to live | 0.21 | 0.03 | 0.40 | 0.01 | 0.39 |
| Urban environment | <–> | Willingness to travel | 0.46 | 0.29 | 0.61 | 0.28 | 0.57 |
| Urban environment | <–> | Physical and mental health | 0.73 | 0.55 | 0.85 | 0.52 | 0.84 |
| Urban economic | <–> | Natural ecology and environment | 0.06 | −0.08 | 0.23 | −0.08 | 0.22 |
| Urban economic | <–> | Willingness to consume | 0.16 | −0.01 | 0.36 | −0.03 | 0.34 |
| Urban economic | <–> | Willingness to live | 0.44 | 0.24 | 0.62 | 0.21 | 0.62 |
| Urban economic | <–> | Willingness to travel | 0.03 | −0.11 | 0.17 | −0.11 | 0.17 |
| Urban economic | <–> | Physical and mental health | 0.12 | −0.05 | 0.32 | −0.07 | 0.30 |
| Natural ecology and environment | <–> | Willingness to consume | 0.27 | 0.14 | 0.42 | 0.14 | 0.42 |
| Natural ecology and environment | <–> | Willingness to live | 0.32 | 0.14 | 0.48 | 0.14 | 0.48 |
| Natural ecology and environment | <–> | Willingness to travel | −0.03 | −0.18 | 0.15 | −0.19 | 0.15 |
| Natural ecology and environment | <–> | Physical and mental health | 0.38 | 0.23 | 0.52 | 0.21 | 0.51 |
| Willingness to consume | <–> | Willingness to live | 0.31 | 0.14 | 0.46 | 0.12 | 0.46 |
| Willingness to consume | <–> | Willingness to travel | 0.37 | 0.20 | 0.51 | 0.20 | 0.51 |
| Willingness to consume | <–> | Physical and mental health | 0.17 | 0.01 | 0.32 | 0.00 | 0.32 |
| Willingness to live | <–> | Willingness to travel | 0.31 | 0.14 | 0.48 | 0.12 | 0.46 |
| Willingness to live | <–> | Physical and mental health | 0.01 | −0.12 | 0.13 | −0.12 | 0.13 |
| Willingness to travel | <–> | Physical and mental health | 0.24 | 0.06 | 0.40 | 0.05 | 0.39 |

*3.5. Data Processing and Analysis*

After the study recovered the predictive questionnaires, the invalid questionnaires were excluded from the statistics of valid questionnaires. The research team went to the local area to collect information and cooperate with the online questionnaire survey platform from June 2021 to October 2021. A total of 1000 questionnaires were distributed, 900 questionnaires (90%) were recovered, and 822 valid questionnaires (91.3%) were obtained at the end. The data were archived with SPSS 26.0 statistical software. First, the basic statistical test method and *t*-test method were used to analyze the public's cogni-

tion of the development status of the urban economy, society, environment, and natural ecological environment. Then, we used AMOS 24.0 statistical software to analyze the correlation between variables. In response to the results of the questionnaire analysis, the semi-structured interview method was used to collect the opinions of 10 respondents with relevant industry and academic backgrounds, as shown in Table 4. Finally, after all the information was integrated, the information was cross-compared with the processes of classification, induction, and sorting, and the multi-check analysis method was used to discuss [51–53].

**Table 4.** Respondent's background information and an overview of the interview outline.

| Code | Identity | Gender | Residence Time/Years of Work Experience | Code | Identity | Gender | Residence Time/Years of Work Experience |
|------|----------|--------|------------------------------------------|------|----------|--------|------------------------------------------|
| P1 | Professor | Male | 11 | R2 | Resident | Female | 58 |
| P2 | Professor | Female | 21 | R3 | Resident | Male | 70 |
| P3 | Professor | Male | 19 | T1 | Tourist | Female | 49 |
| E1 | Entrepreneur | Female | 33 | T2 | Tourist | Male | 59 |
| R1 | Resident | Male | 33 | T3 | Tourist | Female | 37 |

| Construct | Issues |
|-----------|--------|
| Impact of tourism development | Please briefly state your opinion based on the following questions and briefly explain why.<br>1. Has the decision of the water conservancy project changed the current economic, social, environmental, and natural ecological conditions of the community? What are the main reasons?<br>2. After the decision of the water conservancy project is completed, when you live in the local area or engage in leisure tourism activities, has your physical and mental health changed? What are the key factors?<br>3. Are you willing to spend again after you live in the local area or engage in leisure travel activities? What are the key factors?<br>4. When you live in a village, or scenic spot, or engage in leisure tourism activities, are you willing to continue spending and decide to buy land, buy a house, and participate in leisure tourism activities? What are the key factors? |

### 3.6. Ethical Considerations

The research and investigation period and the content of the questionnaire design were anonymous. The research topic, method, question design, respondents' willingness to interview, and the right to use the questionnaire data were clearly explained. Furthermore, the research was conducted with a questionnaire survey method, using statistical software analysis, and there was no physical experiment method in the process.

The research data were rigorous and confidential, and information was obtained after obtaining the respondents' consent. The investigation and sampling process were conducted under the principles of fairness, openness, and impartiality [92,93]. All investigation processes were in line with ethical considerations, so the study did not require the certification of the ethics statement.

### 4. Analysis and Discussion

#### 4.1. Analysis of Background Information

The analysis was based on 822 formal questionnaires. Based on basic statistical analysis, the background information of the sample was as follows: 311 were residents, accounting for 37.834%, and 511 were tourists, accounting for 62.166%. There were 298 males, accounting for 36.253%, and 524 females, accounting for 63.747%. The identity of the sample was mostly tourists, and there were few residents; the gender was mostly female, and there were few males. Based on the above data, the analysis was followed by a statistical test, *t*-test, and structural equation model analysis.

*4.2. The Public's Perception of the Development Status of the Urban Economy, Society, Environment, and Natural Ecological Environment*

First, we analyzed the public's feelings on the economic, social, environmental, and natural ecological development status by basic statistical verification. Then, we used the *t*-test method to analyze the cognition differences of different stakeholders on economic, social, environmental, and natural ecological development. Regarding economic development, improving transportation planning (3.90) was the highest, while developing protection policies and increasing entrepreneurship and employment opportunities (3.90) were the lowest. In terms of social development, the quality of security (3.97) was the highest, and the quality of friendly interaction and tourism services (3.87) was the lowest. In terms of environmental development, more convenient transportation (3.91) was the highest, while landscape and heritage maintenance (3.73) was the lowest. In the natural ecological environment, ecological habitat change (3.72) was the highest, and the number of original ecological species decreased (3.62) was the lowest, as shown in Table 5. It could be seen that the above research results were not consistent with the research Hypotheses 1-1, 1-2, 1-3, and 1-4.

**Table 5.** Analysis of the public's cognition of the development status of urban economy, society, environment, and natural ecological environment.

| Construct | Issues | M | Rank | Identity | | *p* |
|---|---|---|---|---|---|---|
| | | | | **Residents** | **Tourists** | |
| | Combining characteristics with local industries | 3.94 | 2 | 4.10 | 3.84 | 0.002 * |
| Economic | Increase entrepreneurship and employment opportunities | 3.90 | 3 | 3.96 | 3.87 | 0.006 * |
| | Improve transportation planning | 3.95 | 1 | 4.00 | 3.92 | 0.192 |
| | Develop protection policy | 3.90 | 3 | 4.03 | 3.83 | 0.021 |
| | Improve the quality of tourism services | 3.87 | 4 | 4.04 | 3.76 | 0.000 * |
| | Increase leisure opportunities | 3.93 | 3 | 4.06 | 3.85 | 0.000 * |
| Social | Public participation in decision-making | 3.95 | 2 | 4.16 | 3.83 | 0.000 * |
| | Increase friendly interaction | 3.87 | 4 | 3.98 | 3.81 | 0.048 |
| | Increase security quality | 3.97 | 1 | 4.22 | 3.82 | 0.000 * |
| | More convenient transportation | 3.91 | 1 | 4.08 | 3.81 | 0.000 * |
| | Exhaust and noise pollution from automobiles and locomotives increases | 3.86 | 2 | 4.17 | 3.67 | 0.000 * |
| Environment | Adequate trash cans | 3.80 | 3 | 3.83 | 3.79 | 0.097 |
| | Adequate toilets | 3.76 | 4 | 3.95 | 3.65 | 0.001 |
| | Landscape and heritage maintenance | 3.73 | 5 | 3.84 | 3.66 | 0.000 * |
| | Turbid river | 3.64 | 3 | 3.80 | 3.53 | 0.128 |
| Natural ecology and environment | Soil erosion along the river | 3.65 | 2 | 3.85 | 3.50 | 0.008 |
| | Ecological species decline | 3.62 | 4 | 3.75 | 3.52 | 0.074 |
| | Ecological habitat change | 3.72 | 1 | 3.85 | 3.61 | 0.000 * |

* $p < 0.001$.

Then, the differences between residents and tourists in the development of the urban economy, society, environment, and natural ecological environment were analyzed. In terms of characteristics and industry integration, increasing entrepreneurship and employment opportunities, tourism service quality, leisure opportunities, public decision-making participation, security quality, landscape and historical site maintenance, ecological habitat change, and other issues were significantly different ($p < 0.001$), and other issues were not significantly different. Among them, residents were more sensitive to the changes mentioned above.

*4.3. Structural Model Analysis of the Influence of Urban Development Status on Physical and Mental Health, Consumption Willingness, Living and Tourism Willingness, and Behavior*

In this study, the overall model fit was tested regarding the structural model analysis. The overall model fit was tested with seven indicators, including chi-square ($\chi^2$), the ratio of $\chi^2$ to degrees of freedom, RMSEA, CFI, GFI, AGFI, and PCFI [94]. Bagozzi and Yi believed that the smaller the ratio of $\chi^2$ to its degrees of freedom, the better [95,96], and the revised ratio in this study was 5.34. Hair et al. stated that the closer the GFI and AGFI values were to 1, the better [94], and the revised GFI and AGFI in this study were 0.70 and 0.60, respectively. Browne and Cudeck stated that the RMSEA value was less than 0.08, the standard CFI value was greater than 0.90, and PCFI needs to be at least greater than 0.50, while $\geq$0.60 is sufficient and $\geq$0.80 is high [97–99]. When the AGFI, RMSEA, CFI, $\chi^2$, GFI, and other results approach the standard values, it means that they are within the tolerance range [94–99]. This showed that the overall fit index of the results of this study was acceptable, as shown in Table 6.

**Table 6.** Fit analysis of the overall model.

| GFI | Tolerable Range | Correction Mode | Model Fit |
|---|---|---|---|
| $\chi^2$ (Chi-square) | Smaller the better | 2124.57 | |
| $\chi^2/\mathrm{df}$ | <5 | 5.34 | Acceptable |
| GFI | >0.80 | 0.70 | Acceptable |
| AGFI | >0.80 | 0.60 | Acceptable |
| RMSEA | <0.08 | 0.13 | Acceptable |
| CFI | >0.90 | 0.80 | Acceptable |
| PCFI | >0.50 | 0.70 | Pass |

According to Figure 2 and Table 7, the urban economy, environment, natural ecology, environmental development, and physical and mental health cognitive dimensions showed no significance ($p > 0.01$), while urban society and physical and mental health, physical and mental health and consumption willingness, consumption willingness and living willingness, and re-tourism willingness were significant ($p < 0.01$). Therefore, the verification showed that Hypotheses 2, 4 and 5 of this study were not valid, and the others were valid.

**Table 7.** Empirical results of research hypotheses.

| Hypothesis | Relation | Value | Validity |
|---|---|---|---|
| 2 | Urban economic→Physical and mental health | 0.17 | Invalid |
| 3 | Urban social→Physical and mental health | 0.27 * | Valid |
| 4 | Urban environment→Physical and mental health | 0.03 | Invalid |
| 5 | Natural ecology and environment→Physical and mental health | −0.03 | Invalid |
| 6 | Physical and mental health→Willingness to consume | 0.30 * | Valid |
| 7-a | Willingness to consume→Willingness to live | 0.31 * | Valid |
| 7-b | Willingness to consume→Willingness to travel | 0.32 * | Valid |

* $p < 0.05$.

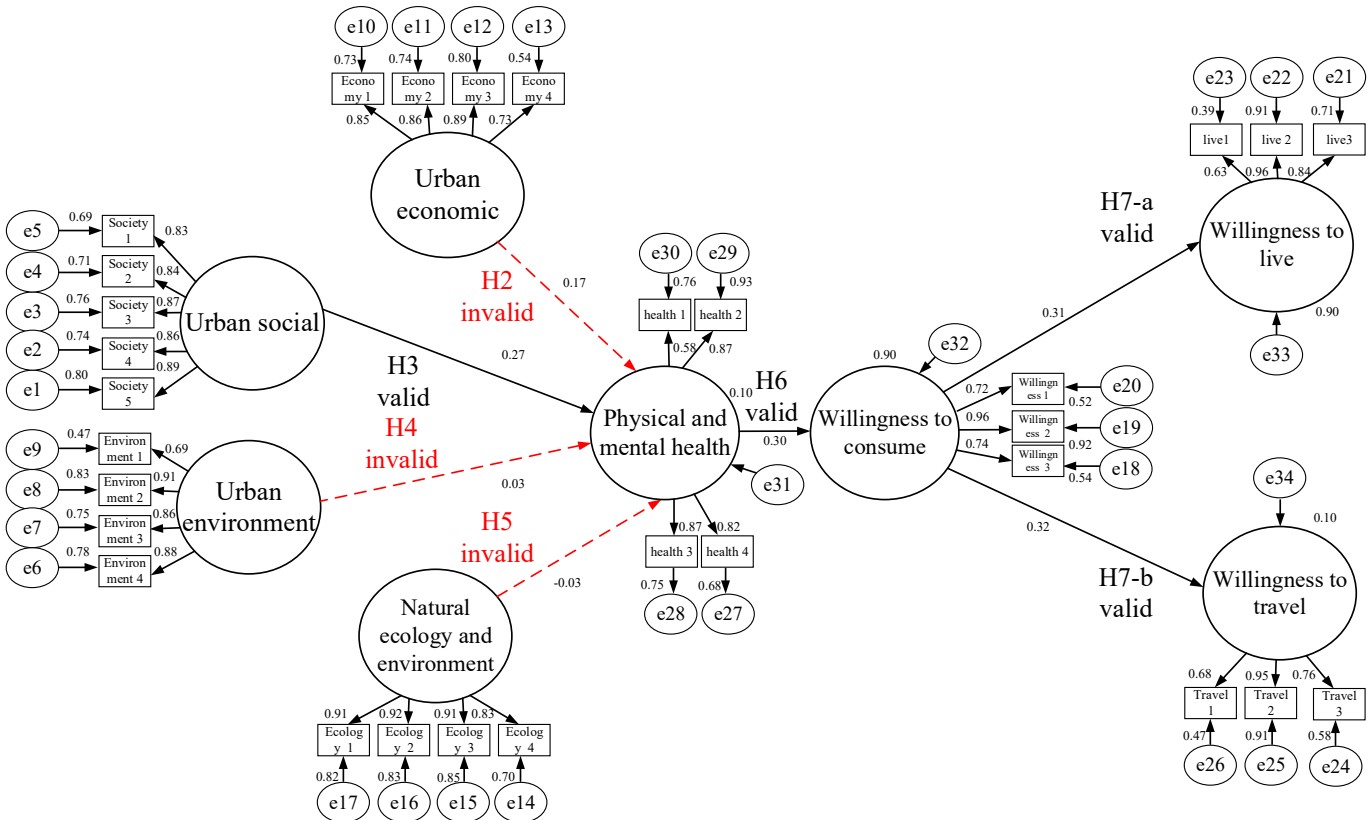

**Figure 2.** Structural pattern analysis of the impact of urban economy, society, environment, natural ecology, and environmental development on physical and mental health, consumption willingness, living, traveling willingness, and behavior.

### 4.4. Discussion

4.4.1. Cognitive Analysis of the Impact of the Three Gorges Dam Water Conservancy Project on Urban Economic, Social, Environmental, and Natural Ecological Development

The study results show that although the Yangtze River is one of the main cultural birthplaces in China, due to the small gap between the river channel and the land along the coast, the river channel is curved [8]. As a result, floods are prone to occur when there is a sudden increase in water flow [7–9]. Floods have occurred regularly for many years, impacting the construction of surrounding villages, endangering people's physical and mental health and safety, and causing heavy losses and casualties [9–11]. The original intention of the Chinese government to build the Three Gorges Dam has improved the flood problem. Later, it can develop a multi-functional water source area due to the advantages of abundant water resources and the diverse natural ecology and environment [13,14].

Therefore, the government has invested a lot of money and manpower to improve urban public construction and transportation planning to facilitate the coming and going of people and to provide large-scale machinery or transportation vehicles. It has increased the convenience for people to engage in leisure, tourism, or business and trade activities, carried out environmental repair or urban renovation work, and provided people with a safe and healthy living, leisure, and tourism environment [13–15].

However, the respondents believe that although the decision-making aimed to improve local predicaments, promote urban development, and improve people's living standards (P1, E1), the current urban industry categories have been highly similar and lacked characteristics (T1, P2). The surrounding environment should be due to the construction of dams and promoting leisure and tourism industries, inundation of historical buildings or monuments, and over-exploitation of existing forest land (P1, P2, P3). People's environmental literacy varied (P2, R1, R3). Excessive waste and exhaust gas from automobiles and motorcycles have damaged the environment and natural ecological environment and

are not conducive to the growth of wild animals and plants (P1, R1-R2). Affected by the epidemic, although the number of tourists has been reduced, the scenic spot has simultaneously laid off staff, resulting in insufficient staff, an inability to meet the needs of tourists (T1-T2), and a decline in service quality. The improper maintenance of landscapes and historical sites has led to a decrease in popularity, affecting tourists' willingness to spend, indirectly reducing job opportunities, and challenging the effectiveness and quality of decision-making (P1-P3, E1, T1-T3).

In addition, the initial purpose of the dam construction has improved flooding, and the surrounding soil and forests are relatively sensitive. So, the scale of urban construction and development of the local government is relatively conservative [6–15]. However, respondents believe that after the dam was built, the city continued to face the threat of flooding, inundating many ancient buildings (P1-P3, R1-R2). With the aging population, there is a serious outflow of youth labor and a shortage of manpower to serve local industries (E1, R1-R3).

Furthermore, although the city is under the construction and development of the Three Gorges Dam, residents believe that public security, transportation planning, and transportation quality have been improved, and transportation convenience has been increased (R1, R3). The combination of characteristics and industries will increase employment, invest in leisure opportunities, and improve the quality of tourism services (R1-R3). However, tourists believe that entrepreneurship and employment opportunities, as well as protection policies, are insufficient (T1-T2). Public interaction is indifferent, and service quality, landscape, and heritage maintenance are poor (T2-T3). At the same time, both residents and tourists believe that the natural ecological status has changed significantly, and the number of species has declined (R1-R3, T1-T3).

Therefore, we believe that the Three Gorges Dam project reduces flooding and provides an environment for villages to promote leisure and tourism activities. However, high-value monuments are inundated, and the land is overdeveloped. The industrial similarity is high, and the number of service personnel is insufficient. People's environmental literacy is poor, vehicle exhaust pollution is serious, and natural environment damage is even more serious. As a result, people's willingness to consume will be affected, the impression of living or traveling will be poor, and the idea of continuing to live or travel will be affected. As a result, only residents believe that local characteristics and industries, entrepreneurship and employment opportunities, tourism service quality, leisure opportunities, public decision-making participation, security quality, and landscape and historical site maintenance have been improved. Still, the problem of ecological habitat destruction is more obvious. As a result, the above research results are inconsistent with the results of the literature [15,16,20,32]. This result verifies that none of the research Hypotheses 1 to 4 are valid.

Therefore, we suggest that first of all, the government should enhance the protection of historical sites and develop historical tourism activities. Second, the government should set up protected areas to reduce the over-exploitation of land and use existing resources to diversify tourism activities or commodities. In addition, the government should improve people's environmental literacy, increase environmental cleaning manpower, reduce garbage, and keep the environment clean and tidy. Next, people should continue to use green energy transportation to reduce vehicle exhaust emissions. Finally, if we can maintain the natural appearance of the city and surrounding ecology, we can increase the positive feelings of tourists and residents.

4.4.2. Analysis of the Impact of Using Existing Resources to Diversify Tourism Activities or Commodities-Urban Economic, Social, Environmental, and Natural Ecological Development Impact on Physical and Mental Health, Willingness to Consume, and Willingness to Live and Travel

Natural-disaster free, well-developed public constructions, a rich ecology, and well-built leisure facilities are the living environment expected by village residents and tourists [15–25]. Although the effect of economic development may affect the quality of life [15–17], the cur-

rent situation of the natural ecological environment may affect people's leisure and tourism experiences [18–23]. The respondents think that even if the economic construction and natural environment are still flawed, if the village can be stabilized in security, people can live a safe and convenient life that can provide the power to stabilize people's minds (P1-P3, R1-R3).

Therefore, we believe that even if the village's economy, environment, and natural ecology are not well developed, as long as community development can provide stable public security, living safety, and convenience in life, it can allow people to maintain normal living patterns and leisure behaviors, stabilize their minds and emotions, and maintain people's physical and mental health. Therefore, only urban society has had an influence on physical and mental health, and the others have had no significant impact. As a result, the above research results are inconsistent with the results of the literature [15–23]; thus, only Hypothesis 3 was confirmed, and the others are not valid.

Therefore, we propose to improve urban public construction, enhance the convenience of life, strengthen the safety of living and tourism, and increase people's trust in urban management. This can increase people's willingness to invest in leisure and travel, reduce anxiety, and maintain physical and mental health.

Secondly, cities or communities are the main living circles of people, and a safe and comfortable living environment can help people reduce the pressure of life [15,16,20,57] and maintain a healthy body and mind. On the other hand, when people's physical and mental health is threatened, they expect to improve their predicament by acquiring resources through consumption [23,71]. Respondents believe that living materials and leisure activities are the keys to improving the quality of life (P1-P2, E1, R1-R2, T1-T3). When people obtain the corresponding resources due to their physical and psychological needs, they can realize the pursuit of better living conditions and quality and achieve the goal of stable physical and mental health (P1-P3).

Therefore, we believe that when people's daily life and the consumption behavior of leisure and tourism activities can meet the individual's physical and psychological needs, the current living conditions and quality can be improved. When people's living conditions and quality are satisfied, life pressure can be eliminated, and psychological burdens can be reduced. It can also generate free time, allowing people to devote themselves to leisure and tourism activities and to further adjust their physical and mental health status. As a result, the above research results are consistent with the results of the literature [15,16,20,23,57,71], and Hypothesis 6 is valid.

Therefore, we suggest that we provide sufficient supplies to meet living needs, create a safe living and leisure environment, and maintain a healthy physical and mental environment. This can provide people with stable leisure activities, reduce anxiety, and maintain physical and mental health.

Furthermore, consumption information or personal experience was the main basis for influencing personal–psychological internalization intentions and transforming actual consumption intentions [60,70]. After personal experience or consumption, when an individual's consumption perception and expected personal physiological or psychological needs reach a balance or exceed the balance [23,57], it would affect the individual to set consumption goals in the next stage [58].

Respondents believe that a safe living space and a good living and development environment can induce people to develop locally (R1-R3). The higher the individual's positive feelings, the better the economic level, and the better the conditions for living and working in peace and contentment (P1-P3). Furthermore, a good tourism experience can promote tourists' willingness to continue to experience (T1-T3).

Therefore, we believe that when people have a good living experience or obtain a positive perception of leisure and tourism experience, it will promote the idea of purchasing land and property or relocating and increase tourists' intention to plan to travel again. As a result, the above research results are consistent with the results of the literature [23,57–60,70], and the assumptions 7-1 and 7-2 are valid.

Therefore, we suggest that cities create a good living environment, plan safe tourism spaces, and increase the public's positive consumption experience and perception. This can prompt the public to have the idea of migrating or planning to travel and consume.

## 5. Conclusions and Suggestions

The study concludes that, although the Three Gorges Dam has successfully reduced the flooding problem, it has also attempted to renovate the local natural ecology and geographical environment and hoped to simultaneously improve the urban economic, social, and environmental development dilemma. Residents believe that the combination of characteristics and industries, entrepreneurship and employment opportunities, tourism service quality, leisure opportunities, public decision-making participation, public security quality, and landscape and historical site maintenance measures have improved. Most people believe that the hidden danger of floods may not be eradicated, the inundated features of historic sites have disappeared, and the forest land has been over-exploited. There are many similarities in industrial categories, the outflow of manpower, the aging of the population, and the lack of grass-roots service manpower. The people's environment and health literacy are different, there is a lot of tourist waste, and air pollution is serious. The current ecological habitat has been significantly changed. Relevant issues may not guarantee a safe and healthy living environment for the public and are not conducive to enhancing the willingness of the public to live or plan to engage in leisure and tourism activities. Moreover, we learned that when the urban social atmosphere is good, and the public construction and living conditions are perfect, the conditions for promoting people's physical and mental health can be achieved. Moreover, when people assess that their physical and mental health is no longer threatened, they will have a willingness to consume to meet their living needs. Finally, when the people obtain sufficient necessities through consumption and meet the basic living conditions, they will increase the idea of traveling or the willingness to live in the local area.

Based on the above description, the study recommends the following.

First, water control decisions need to be re-planed, and emergency escape measures must be formulated. Historical sites should be maintained, and different urban living consumption and leisure and tourism features should be developed. Industrial development blocks should be planned, the public should be guided to learn entrepreneurial technologies, and employment and entrepreneurial opportunities should be increased. This measure can help improve urban development deficiencies.

Second, personal environment and sanitation and epidemic prevention literacy should be improved. Environmental damage and pandemic contagion crises should be reduced. The awareness of decision-making participation should be strengthened. In improving and promoting urban development decisions, the government should be assisted. This measure can help the public improve their decision-making cognition and the identity of the individual and the ecological environment.

Third, based on individual cases, the differences in the development effects and public perceptions of different river blocks and villages should be explored. Then, using other topics such as leisure and environmental literacy, the impact of water conservancy project decision-making on the city, the natural ecological environment, and the people should be discussed. This measure can help fill in the research gaps.

**Author Contributions:** Conceptualization, H.-H.L.; methodology, H.-H.L. and C.-H.H.; software, K.-C.T.; validation, H.-H.L. and C.-H.H.; formal analysis, J.-M.H. and I.-S.C.; investigation, H.-H.L., J.-M.H. and I.-S.C.; resources, K.-C.T.; data curation, K.-C.T.; writing—original draft preparation, C.-H.H.; writing—review and editing, H.-H.L.; visualization, J.-M.H.; supervision, H.-H.L.; project administration, I.-S.C.; funding acquisition, K.-C.T. and I.-S.C. All authors have read and agreed to the published version of the manuscript.

**Funding:** This research received no external funding.

**Institutional Review Board Statement:** The design was in accordance with the scope of human research cases exempted from ethical review by the Ethical Review Board as stipulated in Notice No. 1010265075 of the Department of Health, Executive Yuan, Taiwan. It also complied with Article 1004 and Article 1009 of Title IV of the Chinese Civil Code, which stipulate that scientific research must comply with laws and administrative regulations, not endanger human health, and violate ethics and morality, and protect the safety and health of subjects. All experimental phases were conducted in a fair, impartial, open, and transparent process.

**Informed Consent Statement:** All respondents participated in the study knowingly and authorized the use of the information provided.

**Data Availability Statement:** Not applicable.

**Conflicts of Interest:** All authors have no conflict of interest.

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
