# Peer review of "Influence of Rural Development of River Tourism Resources on Physical and Mental Health and Consumption Willingness in the Context of COVID-19"

_water, doi:10.3390/w14121835_

Round 1

Reviewer 1 Report

Thank you for giving me the possibility to review this manuscript. The paper aims to discuss the influence of river regulation decisions on surrounding villages’ sustainable development. An example of the Three Gorges Dam  used to discuss the influence of river regulation decisions on the sustainable development of surrounding villages. Therefore, the paper will be interesting to readers who are dealing with the issues of sustainable rural development.
The abstract is well structured. The keywords are in line with the terms used in the research.  Arguments and discussion of findings are coherent, balanced, and compelling. The references to esearch results of water resources, tourism development, consumption willingness, and urban development are relevant and mostly up to date.   
The Introduction section correctly puts the research topic in context, with the main purpose of the research clearly stated (lines 143-144). I only would prefer to see a separate section presenting clearly the paper structure.
As to literature review, it is thorough using many relevant sources but, from our point of view, the structure of this section lacks balance. We see two short subsections (20-25 lines) and the huge third subsection of about 100 lines (divided into intro and three sub-subsections).
The authors describe the research framework and research hypotheses, as well as study subjects and limitations. The methodology is appropriate, the authors use mixed research methods, snowball sampling, and convenience sampling to obtain data by means of semi-structured interviews. Research processes and tools are also appropriate.
In the conclusion section, I am not sure that it makes sense to have subsections (5.1 etc), better simply the most important results, implications, research limitations and future research paths. 
The English language and style are not bad but somehow difficult to read due to “foreign” building and excessive length of some phrases. It is recommended to improve the text with the help pf a native speaker.
Overall, the paper is worth publication, with minor revisions proposed above.

Author Response

The Introduction section correctly puts the research topic in context, with the main purpose of the research clearly stated (lines 143-144). I only would prefer to see a separate section presenting clearly the paper structure.

Thanks reviewer:

Thanks to the suggestion, we have revised the narrative and presented a separate description of the main purpose.

As to literature review, it is thorough using many relevant sources but, from our point of view, the structure of this section lacks balance. We see two short subsections (20-25 lines) and the huge third subsection of about 100 lines (divided into intro and three sub-subsections).

Thanks reviewer:

Thanks for the suggestion, we have revised the text description and merged the text description when appropriate.

In the conclusion section, I am not sure that it makes sense to have subsections (5.1 etc), better simply the most important results, implications, research limitations and future research paths. 

Thanks reviewer:

Thanks to the suggestion, we have revised the text description to present the conclusions and future research paths more briefly.

The English language and style are not bad but somehow difficult to read due to “foreign” building and excessive length of some phrases. It is recommended to improve the text with the help pf a native speaker.

Thanks reviewer:

Thanks for the suggestion, we will make a grammar correction.

Overall, the paper is worth publication, with minor revisions proposed above.

Finally, we are very grateful for your support and assistance.

We believe this manuscript will be more visual.

Reviewer 2 Report

Authors should check typos and syntax errors.
Authors should improve the presentation in figure 1 and the relative
text content. 
In figure 1 the demonstrated hypothesis parts and the relative text presented in section 3.2 next hypotheses should be exactly the same and not similar.

'3.2. Research Hypothesis 310
Based on the above inferences and research framework planning, the researcher 311
proposed seven research hypotheses. 312
Hypothesis 1. Assume that the public believes the influences of water conservancy 313
projects on the development of urban economy, society, environment, and natural ecol- 314
ogy are all consistent. 315
Hypothesis 2. Village economic development has a positive and significant impact 316
on physical and mental health cognition. 317
Hypothesis 3. Village social development has a positive and significant impact on 318
physical and mental health cognition. 319
Hypothesis 4. Both natural ecology and environmental development status have a 320
positive and significant impact on physical and mental health cognition. 321
Hypothesis 5. The current status of urban economy, society, environment, natural 322
ecology, and environmental development has a positive and significant impact on phys- 323
ical and mental health cognition. 324
Hypothesis 6. Physical and mental health has a positive and significant impact on 325
willingness to consume. 326
Hypothesis 7. Consumption willingness has a positive and significant impact on 327
both living and travel willingness.'

Therefore, the above text should  be edited properly between lines 311 and 328.

According to section 3.2 should exist a
better presentation between figures linkage with text and results  for
  local residents and people with local tourism  
experience as the object.
Authors should present better the text in results analysis  in terms of uncertainty that they found as it is described in lines 347, 348.
 Although 347 there were restrictions on the sampling conditions of the online questionnaire, the un-348 certainty of the respondents' willingness and identity was high.

According to authors analysis about the PCFI index presented in line 492
'least greater than 0.50 [97].' more datasets may be presented so as to improve its analysis.
Table 7 should be edited properly.
In lines 494 - 497 is said 
'It 494
showed that the overall fit index of the results of this study was acceptable, as shown in 495
Table 7. 496
497' but in table 7 last line demonstrates that for PCFI as it has been presented 'PCFI >0.50 0.70 pass' it is not acceptable but pass.

Authors should improve presentation, typos and syntax errors in English language between presented figures / tables - results and 
linkage of them in text. 
References should be updated properly and present better the results in relation to reference in literature findings, with better explanations for  tourism development improvement in future. 
Authors should improve relative indicative content for next  found text in similar sentences respectively found in article.
'As a result, the above research results are in-575 consistent with the results of the literature [15-16, 20, 32]. This result verifies that none of 576 the research hypotheses 1 to 4 are valid.' lines 575-576.
'As a result, the above research results are consistent with the results of 612 the literature [15-16, 20, 23, 57, 71], and hypothesis 6 is valid.' lines 612-613.
In conclusions solutions should be presented about environmental pollution based on next content of sentences found in article.

'The  people's environment and health literacy are different, there is a lot of tourist waste, and 643 air pollution is serious.'line 642.

'It is also found that the effectiveness of urban so-cial development is the key to influencing people's perception of physical and mental health, influencing their willingness to consume, and changing their intention to live or travel.'lines 647 - 649.

Better presentation in results sections is needed in relation to the findings from the working hypotheses study.

Author Response

Authors should improve the presentation in figure 1 and the relative text content.

In figure 1 the demonstrated hypothesis parts and the relative text presented in section 3.2 next hypotheses should be exactly the same and not similar.

Thanks reviewer:

Thanks to your suggestion, we have revised Figure 1 along with the content; and we have supplemented the hypothetical narrative to match the narrative of Figure 1.

'3.2. Research Hypothesis 310

Based on the above inferences and research framework planning, the researcher proposed seven research hypotheses. Hypothesis 1. Assume that the public believes the influences of water conservancy projects on the development of urban economy, society, environment, and natural ecology are all consistent.

Hypothesis

  1. Village economic development has a positive and significant impact on physical and mental health cognition.

Hypothesis 3. Village social development has a positive and significant impact on physical and mental health cognition.

Hypothesis 4. Both natural ecology and environmental development status have a positive and significant impact on physical and mental health cognition.

Hypothesis 5. The current status of urban economy, society, environment, natural ecology, and environmental development has a positive and significant impact on physical and mental health cognition.

Hypothesis 6. Physical and mental health has a positive and significant impact on willingness to consume.

Hypothesis 7. Consumption willingness has a positive and significant impact on both living and travel willingness.

Therefore, the above text should  be edited properly between lines 311 and 328.

Thanks reviewer:

Thanks for your suggestion, we've added additional instructions.

According to section 3.2 should exist a better presentation between figures linkage with text and results  for  local residents and people with local tourism experience as the object.

Authors should present better the text in results analysis  in terms of uncertainty that they found as it is described in lines 347, 348.

Although there were restrictions on the sampling conditions of the online questionnaire, the un certainty of the respondents' willingness and identity was high.

Thanks reviewer:

Thanks for your suggestion, we have revised this description. "Although the sampling conditions of the online questionnaire are limited, the respondents' willingness to answer and actual travel experience are highly uncertain."

According to authors analysis about the PCFI index presented in line 492 'least greater than 0.50 [97].' more datasets may be presented so as to improve its analysis.

Table 7 should be edited properly.

In lines 494 - 497 is said 'It showed that the overall fit index of the results of this study was acceptable, as shown in Table 7. 496

497' but in table 7 last line demonstrates that for PCFI as it has been presented 'PCFI >0.50 0.70 pass' it is not acceptable but pass.

Thanks reviewer:

Dear reviewer, thank you for the reminder.

We rely on the arguments of the following literature that PCFI needs to be at least greater than 0.50, while ≥ 0.60 is sufficient and ≥ 0.80 is high.

Hair, J.F., Anderson, R.E., Tatham, R.L., and Black, W.C. Multivariate data analysis. Prentice-Hall, 1998.

Byrne BM. Structural equation modeling with Amos: basic concepts, applications, and programming. 3rd ed. New York: Routledge; 2016.

Browne, M. W., & Cudeck, R. (1993). Alternative ways of assessing model fit. In K. A. Bollen and J. S. Long (Eds.), Testing structural equation models, Newbury Park, CA: Sage.

Authors should improve presentation, typos and syntax errors in English language between presented figures / tables - results and inkage of them in text.

Thanks reviewer:

Thanks for the suggestion, we will fix the English spelling in the graphs and tables.

References should be updated properly and present better the results in relation to reference in literature findings, with better explanations for  tourism development improvement in future.

Thanks reviewer:

References should be appropriately updated and better present results related to literature research findings to better explain future improvements in tourism development.

Authors should improve relative indicative content for next  found text in similar sentences respectively found in article.

'As a result, the above research results are in-575 consistent with the results of the literature [15-16, 20, 32]. This result verifies that none of 576 the research hypotheses 1 to 4 are valid.' lines 575-576.

'As a result, the above research results are consistent with the results of 612 the literature [15-16, 20, 23, 57, 71], and hypothesis 6 is valid.' lines 612-613.

In conclusions solutions should be presented about environmental pollution based on next content of sentences found in article.

'The  people's environment and health literacy are different, there is a lot of tourist waste, and 643 air pollution is serious.'line 642.

Thanks reviewer:

Thanks for the suggestion, we will supplement the suggestions for Hypothesis 1 and Hypothesis 6 in response to the above suggestions. And put forward the improvement plan related to the environment. Such as the red font in 4.3.1 and 4.3.2.

'It is also found that the effectiveness of urban so-cial development is the key to influencing people's perception of physical and mental health, influencing their willingness to consume, and changing their intention to live or travel.'lines 647 - 649.

Thanks reviewer:

Thanks for the suggestions, we will supplement the suggestions for the above two conclusions. Such as 4.3.1, the last paragraph is in red font.

Better presentation in results sections is needed in relation to the findings from the working hypotheses study.

Thanks reviewer:

Thanks for the suggestion, based on the analysis results, we will strengthen the narrative and present the influence of the current urban development, physical and mental health, consumption intention, residence and tourism intention.

Finally, we are very grateful for your support and assistance.

We believe this manuscript will be more visual.

Reviewer 3 Report

Dear authors,

The article contains valuable information for the stakeholders in terms of rural development. Please find below few sugestions to improve the manuscript:

  • please be specific: ”the researcher took a lot of time on this study” (L345)
  • please specify the answer ratio (L374)
  • please specify the limits of the research

All the best!

Author Response

The article contains valuable information for the stakeholders in terms of rural development. Please find below few sugestions to improve the manuscript:

    please be specific: ”the researcher took a lot of time on this study” (L345)

Thanks reviewer:

Thanks for your suggestion, which we mentioned in the descriptions of "3.3.1. Exploratory factor analysis" and "3.4. Data processing and analysis".

    please specify the answer ratio (L374)

Thanks reviewer:

We have added a description of the effective recovery rate of the pretest questionnaire.

    please specify the limits of the research

Thanks reviewer:

We have supplemented the relevant instructions. "3.2. Study subjects, limitations, and sampling methods"

Round 2

Reviewer 2 Report

Check typos and syntax errors. Although the requested parts of corrections have been made, the authors should check the better presentation in figures , tables and linkage the results within text.

Also for the next references found in internet the plagiarism check in text should be checked and properly improved so as to be less than 2 % that was found.

1 https://www.researchgate.net/publication/287645318_The_role_and_impact_of_tourism_on_local_economic_dev

elopment_A_comparative_study

INTERNET

2%

2 https://pubmed.ncbi.nlm.nih.gov/35013692/

INTERNET

2%

Author Response

Dear Review

We thank you and would like to take the time to provide suggestions for improving the manuscript.
We believe these suggestions have led to better outcomes for the manuscript.

During this revision, we have adjusted the description and wording of the manuscript to avoid similarity with other literature.

Once again, we thank you for your assistance and wish all the best in the future.